# Transparency in conducting and reporting research: A survey of authors, reviewers, and editors across scholarly disciplines

Mario Malički[1]*, IJsbrand Jan Aalbersberg[2], Lex Bouter[3,4], Adrian Mulligan[5], Gerben ter Riet[1,6]

1 Urban Vitality Centre of Expertise, Amsterdam University of Applied Sciences, Amsterdam, The Netherlands, 2 Elsevier, Amsterdam, The Netherlands, 3 Faculty of Humanities, Department of Philosophy, Vrije Universiteit, Amsterdam, The Netherlands, 4 Department of Epidemiology and Data Science, Amsterdam University Medical Centers, Amsterdam, The Netherlands, 5 Elsevier, Oxford, United Kingdom, 6 Department of Cardiology, Amsterdam University Medical Centers, Amsterdam, The Netherlands

* mario.malicki@mefst.hr

**Data Availability Statement:** Study registration and protocol, survey questions, survey invites, statistical analysis codes, anonymised data, and paper and appendix tables are available on our

## Abstract

Calls have been made for improving transparency in conducting and reporting research, improving work climates, and preventing detrimental research practices. To assess attitudes and practices regarding these topics, we sent a survey to authors, reviewers, and editors. We received 3,659 (4.9%) responses out of 74,749 delivered emails. We found no significant differences between authors', reviewers', and editors' attitudes towards transparency in conducting and reporting research, or towards their perceptions of work climates. Undeserved authorship was perceived by all groups as the most prevalent detrimental research practice, while fabrication, falsification, plagiarism, and not citing prior relevant research, were seen as more prevalent by editors than authors or reviewers. Overall, 20% of respondents admitted sacrificing the quality of their publications for quantity, and 14% reported that funders interfered in their study design or reporting. While survey respondents came from 126 different countries, due to the survey's overall low response rate our results might not necessarily be generalizable. Nevertheless, results indicate that greater involvement of all stakeholders is needed to align actual practices with current recommendations.

## Introduction

Scholarly publishing has been steadily growing for the last two centuries with estimates of more than 3 million articles published per year [1]. In recent times, there has been an increasing focus on the detection and prevention of misconduct and detrimental research practices, in enhancing research rigor and transparency, and in cultivating a research climate best suited to foster these goals [2–5]. Specifically, calls have been made to increase the transparency in conducting and reporting research by registering projects and data analyses plans before data collection, using reporting guidelines when writing up studies, posting preprints, sharing (raw) data, reproducing or replicating studies, as well as rewarding, hiring or promoting

project's data repository: http://dx.doi.org/10.17632/53cskwwpdn.6.

**Funding:** This study was part of an Elsevier funded project: Fostering Transparent and Responsible Conduct of Research: What can Journals do?. Details of the project are available on our project's data repository: http://dx.doi.org/10.17632/53cskwwpdn.6. The funder (other than the funder-affiliated authors IJJA and AM) had no role in study design, data collection and analysis, decision to publish, or preparation of the manuscript.

**Competing interests:** IJsbrand Jan Aalbersberg is Senior Vice President of Research Integrity at Elsevier, and Adrian Mulligan is a Research Director for Customer Insights at Elsevier. Mario Malicki is a Co-Editor-In-Chief or Research Integrity and Peer Review journal. Other authors declare no competing interests. This does not alter our adherence to PLOS ONE policies on sharing data and materials.

researchers based on these practices [6–9]. One such call was made in 2014, when the Transparency and Openness Promotion (TOP) Committee developed eight transparency standards related to: 1) citations, 2) data, 3) analytic methods (code), 4) research materials, 5) data and analysis reporting, 6) preregistration of studies, 7) preregistration of analysis plans, and 8) replication of research [10]. Since then, more than 5,000 journals and 80 organizations became TOP signatories [11]. However, to the best of our knowledge, attitudes related to all aspects of TOP guidelines have not been systematically assessed, and, yet, without agreement from all stakeholders regarding these guidelines, it is unlikely that scholarly practices will change.

It was therefore our goal to assess differences in attitudes and perceptions between authors, reviewers, and editors regarding the TOP guidelines, differences in perceptions of their work climates, and differences in their perceived prevalence of responsible and detrimental research practices.

## Materials and methods

We reported our study following the Strengthening the Reporting of Observational Studies in Epidemiology (STROBE) guidelines [12], as well as the Checklist for Reporting Results of Internet E-Surveys (CHERRIES) [13].

### Study design and participants

Full methodological details of our study, including its registration and study protocol are available at our project's data repository [14]. In short, we sent a survey to 100,000 email addresses. The email addresses were from: 1) randomly selected corresponding authors of papers indexed in Scopus (n = 99,708); or 2) editors whose *Instructions to Authors* we analysed in our previous study (n = 292) [15].

### Setting

The survey was sent on 24 April 2018, had reminders on 9 and 24 May, and was closed on 12 June of that same year. The survey invitation and reminders, full survey questions and details of their development and testing are available on our project's data repository [14]. Respondents could skip items and change the answers they gave until submitting their responses by clicking on the *"close the survey"* button. Estimated time to finish the survey was 12 minutes (based on pilot results), and was listed in the survey invite letter.

### Variables and their measurement

The survey was divided into 4 sections and the questions were presented in the same order to all participants:

1. attitudes towards transparency in conducting and reporting research (11 questions covering the TOP guidelines with 5-point Likert-type answers ranging from strongly agree to strongly disagree);

2. perceptions of work climate (13 questions with 5-point Likert-type answers ranging from strongly agree to strongly disagree);

3. perceived prevalence of responsible and detrimental research practices (14 questions with 5-point Likert-type answers ranging from very prevalent to never occurring);

4. sociodemographic and knowledge of statistics questions (10 questions with categorical answers).

All questions for the first three sections also included a *Don't know/Not applicable* option. In section 4, respondents could select one of 28 prespecified scholarly fields, a multidisciplinary category or the "*other*" category where they could write their field(s) using an open text response format. All answers they provided in free text, as well as the 29 choices, were recoded to one of the 6 major categories that were used in our previous study on *Instruction to Authors*: *Arts & Humanities*, *Health Sciences*, *Life Sciences*, *Physical Sciences*, *Social Sciences*, and *Multi-disciplina*ry [15]. Additionally, we had open-ended questions (free text format) that explored reasons for respondents' (dis)agreement with questions of first three sections, and a final survey question for general feedback on the survey. In order to maintain the focus on quantitative data and have a reasonable reporting length, analysis of open-ended answers is planned for another publication.

## Statistical methods and study bias

We conducted all analyses in STATA v.13, with statistical code and outputs published on our project's data repository [14]. The grouping of respondents to authors (A), reviewers (R) and editors (E) is explained in detail in the Appendix. The main outcomes (answers to questions in sections 1 to 3 listed above) are presented as absolute numbers and percentages (calculated on the basis of the number of respondents that answered a specific question). For readability, percentages are shown only for those who agreed or strongly agreed to statements, or that perceived prevalent or very prevalent practices. Data for all answer options are available on our project's data repository [14]. Differences in sociodemographic characteristics and knowledge of statistics between authors, reviewers and editors were explored using the chi-squared test for categorical variables, and Kruskal-Wallis test for respondents' age. To explore possible associations between answers to questions of sections 1 to 3 and explanatory variables (socio-demographic characteristics and knowledge of statistics) we used ordinal regression analyses. We conducted regressions for all 38 questions individually. We also explored treating questions of each section as separate scales (with the scales consisting of 11, 13 and 14 questions, respectively). We constructed 3 summary scores for those scales, which we then also explored in ordinal regression analyses. To adjust for multiple comparisons, we considered $p \leq 0.001$ as statistically significant (based on the Bonferroni correction method of dividing 0.05 by 50, which was the number of conducted regressions rounded-up to the nearest decile). For readability, regression outputs are presented graphically in the Appendix, while details of analyses, odds ratios and their associated 95% CI are available on our project's data repository [14].

## Respondents' inquiries and deviations from the protocol

Due to miscommunication within the team, instead of adding the collected email addresses of the editors from our previous study on *Instruction to Authors [15]* to the total of 100,000 planned invites, 58 of collected email addresses were used for piloting the survey, and 292 were incorporated into the 100,000 invites sent. Additionally, during survey creation, instead of planned options of 0, 1–4, 5–10,10+ for question—*How many articles (approximately) did you review in the last 12 months*? survey options were instead 1, 2–5, 6–10, and 10+. In the invite email, respondents were asked to contact us if they encountered any (technical) difficulties. Details of their questions are listed in the Appendix.

## Ethics approval

An ethics waiver for the study has been obtained on 6 April 2018 from the Medical Ethics Review Committee of the Academic Medical Center, University of Amsterdam (reference number W18_112). The survey invitation (available on our project's data repository [14])

included information on the study purpose, investigators, estimated length to complete the survey, planned sharing of anonymised data, and publication of summary results. No incentives were offered for participation.

## Results

Our overall response rate was 4.9% (3,659 out of 74,749 delivered emails) and included responses from 1,389 authors, 1,833 reviewers, and 434 editors. Respondents came from 126 countries, most commonly USA (16%), India (8%), Italy (7%), and Brazil (5%). Respondents' median age was 44 (IQR 35 to 55, range 23 to 90). The majority worked for universities or colleges (62%), and were male (66%). While respondents came from all scholarly disciplines, most were from *Physical* (33%) and *Life Sciences* (25%), followed by *Multidisciplinary* (18%) and *Health Sciences* (13%). Similarly, while respondents came from all career stages, most had a publication record of 6 to 25 publications (39%), or more than 50 publications (25%). Finally, most respondents considered themselves to have either basic (44%) or intermediate (45%) knowledge of statistics. Full details on response rate calculation, assignment of respondents to authors, reviewers, and editors, their self-declared statistical knowledge, and sociodemographic characteristics are available in the Appendix. Summary results are presented below, while all answers, as well as percentages of respondents who chose "*Don't know*" or "*Not applicable*" options, are shown in the Appendix.

### Attitudes toward transparency in reporting and conducting research

Respondents' attitudes towards 11 statements on transparency in reporting and conducting research, which were based on TOP guidelines, are shown in Fig 1. The lowest support, across all respondents, was for preregistering studies prior to conducting the research (21%), and the highest for appropriately citing all data, analytic methods (program code) and materials used in the study (95%). Regression analyses (Appendix Table A7 in S1 Appendix) revealed no significant differences between authors, editors and reviewers and significant associations for 3 factors: 1) discipline–*Health Sciences* researchers had more positive attitudes towards transparency in reporting and conducting research than researchers of other disciplines; 2) number of publications–researchers with less than 6 publications had more positive attitudes than other researchers; 3) country–respondents from India had more positive attitudes than respondents from other countries (Fig 2).

### Perceptions of work climate

Respondents' perceptions on 13 statements about their own work climate are shown in Fig 3. Most respondents stated that they share their research data with other researchers unless legal or ethical reasons prevent them from doing so (79%), and that having access to others' data benefits (or would benefit) them (79%). Two thirds (66%) considered the quality of peer review they received to be generally high, as well as the quality of publications in their field (64%). One fifth of respondents (20%) stated that due to the pressure to publish they sacrifice the quality of their publications for quantity, and one seventh of respondents (14%) stated that funders or sponsors interfered in their study design or study reporting. Regression analyses (Appendix Table A8 in S1 Appendix) revealed no significant differences between authors, editors and reviewers and significant associations with perceptions of work climate with two factors: 1) country–respondents from India or USA had more positive perceptions of their work climate than respondents from other countries; 2) age–younger respondents perceived a worse work climate than older respondents (e.g., younger respondents perceived more pressure to

| Statement - % of respondents who (strongly) agreed* | Authors (n=1,389) | Reviewers (n=1,833) | Editors (n=434) | Graphical comparison | Total (n=3,659) |
|---|---|---|---|---|---|
| Authors must appropriately cite all data, analytic methods (program code) and materials used in the study. | 93 | 96 | 95 | | 95 |
| Authors must indicate whether the data, analytic methods (program code), and research materials will be made available to any researcher for purposes of reproducing the results or replicating the procedures. | 82 | 85 | 83 | | 83 |
| Authors must follow appropriate reporting guidelines (e.g. those from www.equator-network.org) for disclosing key aspects of the research design and data analysis. | 75 | 74 | 71 | | 74 |
| Journals must check and enforce appropriate reporting guidelines (e.g. those from www.equator-network.org) for disclosing key aspects of the research design and data analysis. | 71 | 67 | 62 | | 68 |
| Journals must encourage submission of replication studies, particularly of research they publish. | 59 | 64 | 57 | | 61 |
| Authors must deposit all data, analytic methods (program code), and research materials to a trusted repository. All exceptions due to legal or ethical reasons must be identified at article submission. | 62 | 60 | 55 | | 60 |
| Journals must verify that the findings are replicable using the deposited authors' data and methods of analysis. | 58 | 46 | 38 | | 50 |
| Journals must employ a two-stage review process for replication studies - in the first stage review the proposal for the replication study, and in the second the full conducted study. | 42 | 33 | 30 | | 36 |
| Journals must check and indicate within the publication that a study had been preregistered prior to the research being conducted. | 29 | 22 | 19 | | 24 |
| Authors must include the full data analysis plan in their study preregistration. | 26 | 21 | 21 | | 23 |
| Authors must preregister their study prior to conducting the research. | 26 | 19 | 18 | | 21 |

**Fig 1. Respondents' attitudes toward transparency in reporting and conducting research.** In the numerical comparison differences between groups larger than 5% are in bold. In the graphical comparison highest percentage is darker. * Questions are sorted by the total agreement percentage. Order of questions as they were asked in the survey is presented in the Appendix. Slight variations exist for the number of respondents per question, exact numbers are available on our project's data repository.

sacrifice quality for quantity, and they more often stated that long peer review times had negatively impacted their careers, Fig 2).

## Perceived prevalence of responsible and detrimental research practices

Respondents' perceived prevalence of responsible and detrimental research practices are shown in Fig 4. Among detrimental practices, 38% of respondents perceived undeserved authorship, and 33%, prior relevant research not being cited, as (very) prevalent in their field. Of responsible practices, 32% perceived self-reporting of limitations, 19% perceived sharing of raw data, and 6% publication of studies with null or negative results to be (very) prevalent. Regression analyses (Appendix Table A9 in S1 Appendix) revealed editors perceiving fabrication, falsification, plagiarism, and omitting of references to be more prevalent than authors or reviewers. Additionally, editors also perceived undeclared conflicts of interest, and publication of studies with null or negative results to be more prevalent than reviewers (but not than authors, Appendix Table A9 in S1 Appendix).

Regression analyses also revealed 3 other factors associated with the prevalence of research practices: 1) country–respondents from India or USA perceived detrimental research practices to be more prevalent than respondents from other countries; 2) discipline–*Health Sciences* respondents perceived responsible practices, i.e., use of reporting guidelines, open peer review, publication of studies with null or negative results, and reporting of study limitations as more prevalent than respondents from other disciplines, but they also perceived one detrimental

| Statement / practice | Comparison groups | | Graphical comparison |
|---|---|---|---|
| **Examples of attitudes toward transparency in reporting and conducting research** | | | |
| Journals must verify that the findings are replicable using the deposited authors' data and methods of analysis. | India | Other Countries | |
| | 69 | 46 | |
| Authors must preregister their study prior to conducting the research. | Health Sciences | Other Sciences | |
| | 39 | 17 | |
| Journals must employ a two-stage review process for replication studies - in the first stage review the proposal for the replication study, and in the second the full conducted study. | Authors with <6 publications | Authors with ≥6 publications | |
| | 46 | 33 | |
| **Examples of perceptions of work climate** | | | |
| Quality of peer review I received for my publications was generally high. | India | Other Countries | |
| | 79 | 65 | |
| I am willing to publish studies with null or negative results. | USA | Other Countries | |
| | 76 | 57 | |
| Due to the pressure to publish, I sacrifice the quality of my publications for quantity. | Respondents <35 years old | Respondents ≥35 years old | |
| | 28 | 18 | |
| **Examples of perceptions of prevalence of (detrimental) research practices** | | | |
| Plagiarism | India | Other Countries | |
| | 40 | 10 | |
| Self-reporting of study limitations | Health Sciences | Other Sciences | |
| | 55 | 28 | |
| Undeserved authorship (i.e., guest or gift authorship) | Respondents <35 years old | Respondents ≥35 years old | |
| | 46 | 37 | |

**Fig 2. Examples of univariate comparisons that were confirmed in regression analyses to be significantly associated with attitudes towards transparency in reporting and conducting research, perceptions of work climate or perceived prevalence of responsible and detrimental research practices.** In the graphical comparison highest percentage is darker. Slight variations exist for the number of respondents per question, exact numbers are available on our project's data repository.

practice—ghost writing—to be more prevalent than the respondents from other disciplines; 3) age–younger respondents perceived undeserved authorship, as well as prior relevant research not being cited, as more prevalent than older respondents (Fig 2).

| Statement - % of respondents who (strongly) agreed* | Authors (n=1,389) | Reviewers (n=1,833) | Editors (n=434) | Graphical comparison | Total (n=3,659) |
|---|---|---|---|---|---|
| Unless legal or ethical reasons prevent it, I share my research data with other researchers. | 75 | 81 | 81 | | 79 |
| Having access to others' research data benefits/would benefit my own research. | 77 | 81 | 76 | | 79 |
| It is difficult to publish studies with null or negative results. | 69 | 76 | 70 | | 73 |
| Quality of peer review I received for my publications was generally high. | 67 | 66 | 63 | | 66 |
| Quality of publications in my field is generally high. | 66 | 62 | 62 | | 64 |
| I am willing to publish studies with null or negative results. | 54 | 61 | 62 | | 59 |
| Authors recommending peer reviewers for their research is a good practice. | 57 | 51 | 55 | | 54 |
| Quality of mentoring of young scientists/PhD students in my field is generally high. | 49 | 55 | 57 | | 53 |
| I find it easy to obtain ethics opinions (approvals) for my studies. | 45 | 46 | 49 | | 46 |
| Time taken to have my work peer-reviewed has affected my career negatively. | 27 | 23 | 20 | | 24 |
| There is sufficient funding availability for research in my field. | 22 | 21 | 20 | | 21 |
| Due to the pressure to publish, I sacrifice the quality of my publications for quantity. | 24 | 19 | 15 | | 20 |
| Funders/sponsors interfere in my study design or study reporting. | 16 | 13 | 12 | | 14 |

**Fig 3. Respondents' perceptions toward their work climate.** In the numerical comparison differences between groups larger than 5% are in bold. In the graphical comparison highest percentage is darker. * Questions are sorted by the total agreement percentage. Order of questions as they were asked in the survey is presented in the Appendix. Slight variations exist for the number of respondents per question, exact numbers are available on our project's data repository.

## Discussion

Our study has shown that authors, reviewers and editors were not supportive of all TOP recommendations for transparency in conducting and reporting of research. For example, while 95% of respondents (strongly) agreed that researchers must appropriately cite study data, methods and materials; a large majority (74%) that authors must follow reporting guidelines, that journals must encourage publication of replication studies (61%), or that authors must share data (60%); only half (50%) felt that journals have to verify that study findings are replicable using the deposited authors' data and methods of analysis, and 21% that studies must be preregistered. While we found no significant differences in these attitudes between authors, reviewers, and editors, we did observe differences between respondents of different countries, disciplines and research seniority. Overall, younger respondents, those from *Health Sciences*, or from India, had more positive attitudes towards the TOP recommendations. Direct comparisons of our results with other surveys are difficult, as to the best of our knowledge, no surveys have addressed all aspects of TOP guidelines, nor surveyed respondents from all disciplines or countries represented in our survey since the TOP guidelines were published. Furthermore, year when the survey is conducted, wording differences, use of scales versus single questions to assess attitudes, as well as differences in collected sociodemographic data that

| Practice - % of respondents who perceived it as (very) prevalent* | Authors (n=1,389) | Reviewers (n=1,833) | Editors (n=434) | Graphical comparison | Total (n=3,659) |
|---|---|---|---|---|---|
| Undeserved authorship (i.e. guest or gift authorship) | **34** | **40** | 39 | | 38 |
| References being omitted (i.e. prior relevant research not being cited) | **27** | **35** | 46 | | 33 |
| Self-reporting of study limitations | **28** | 33 | 35 | | 32 |
| Use of reporting guidelines for disclosing key aspects of the research design and data analysis | 24 | 25 | 26 | | 25 |
| Sharing of relevant raw data underlying a research study | 17 | 21 | 20 | | 19 |
| Publishing of authors' versions (non-peer reviewed versions) on pre-print servers | 17 | 20 | 18 | | 19 |
| Ghost writing (i.e. author(s) not being acknowledged) | 15 | 13 | 15 | | 14 |
| Undeclared conflict(s) of interest / competing interest(s) | 17 | 12 | 14 | | 14 |
| Plagiarism | **16** | **10** | 15 | | 13 |
| Open peer review (i.e. reviewers signing their review reports) | 16 | 11 | 12 | | 13 |
| Fabrication or falsification (incl. Image manipulation) | 12 | 9 | 12 | | 11 |
| Publication of corrections (i.e. errata, corrigenda) | 11 | 10 | 13 | | 11 |
| Publication of studies with null or negative results | 8 | 5 | 5 | | 6 |
| Publication of retractions (i.e. study withdrawal) | 6 | 4 | 4 | | 5 |

**Fig 4. Respondents' perceptions of perceived prevalence of responsible and detrimental research practices.** In the numerical comparison differences between groups larger than 5% are in bold. In the graphical comparison highest percentage is darker. * Questions are sorted by the total agreement percentage. Order of questions as they were asked in the survey is presented in the Appendix. Slight variations exist for the number of respondents per question, exact numbers are available on our project's data repository.

might be explored as explanatory variables, often don't allow direct comparisons [16]. Nevertheless, a survey of 1,015 authors of observational studies, also conducted in 2018, showed that 63% of respondents used reporting guidelines, and that their attitudes towards, awareness, and use of reporting guidelines are influenced by journals' endorsements [17], which was also echoed in earlier studies [18]. Our previous analysis of journals' endorsements, however, showed that only 13% of journals across disciplines recommended the use of reporting guidelines, and only 2% required it [15]. Data sharing practices across sciences have not been systematically explored, but recent estimates indicated that data sharing was mentioned in 15% of biomedical [19], and in only 2% of psychological articles [20]. A 2020 systematic review indicated that most researchers have positive attitudes toward data sharing [21]. This discrepancy of positive attitudes versus lack of actual practice of sharing data is influenced by many factors, including requirements for job selection and promotion, dedicated funding and skills, as well as incentives, time and training required to prepare (anonymised) data for sharing [21]. Regarding preregistration, a 2017 survey of 275 authors of systematic reviews or meta-analyses, showed

that 37% of participants agreed with making protocols mandatory [22]. While that percentage was higher than in our study, the study's sample size was smaller, respondents were predominantly from biomedical disciplines, and the questions only pertained to preregistration of systematic reviews.

It is likely that the lack of support of editors in our study towards all aspects of TOP recommendations is one of the contributing factors why these recommendations have not been endorsed by a larger number of journals today. The roles of editors and journals in endorsing or requiring specific practices, the lack of resources they often face, and the lack of proper intervention studies, were discussed extensively in our previous publications [15,23]. Here we would like to add that uptake and attitudes toward different practices are also likely influenced by whom and how recommendations are made. More rigorous methodological steps during recommendation development (similar to those for reporting guidelines or clinical practice guidelines), and open feedback calls might have led to higher initial uptakes. Additionally, published case reports with cost estimates and practical tips from those involved in recommendation development, that then managed to change practices of their journals, departments or institutions, could perhaps lead to additional endorsements.

Our study has also shown that work climates of authors, reviewers, and editors still had room for significant improvements. Approximately two thirds of respondents (66%) found the quality of peer review they received, as well as the quality of publications in their field (64%) to be generally high, one fifth of the respondents (20%) stated that due to the pressure to publish they sacrifice the quality of their publications for quantity, and 14% stated that funders or sponsors interfere in their study design or study reporting. The finding regarding pressure to publish was associated with respondent's age, with younger respondents feeling more pressure to sacrifice quality for quantity. Younger respondents also felt that long peer review times had a more negative effect on their careers than did the other researchers. Again, direct comparison with previous studies is difficult due to differences in questions, but a recent survey on 7,670 postdoctoral researchers from 93 countries indicated that 56% had a negative outlook on their careers, and that only 46% would recommend to their younger selves to pursue a career in research [24]. Furthermore, as most postdocs in that survey reported being hired for only short periods of time, it is likely that long peer review times, and the number of publications might have more impact on their job prospects than of (tenured) academics. The overall high satisfaction of researchers with peer review we found, has also been reported in previous studies, with a caveat of known differences in reported satisfaction among those that had their papers rejected vs those that had them accepted [25].

Finally, our study showed that most commonly perceived detrimental research practices were undeserved authorship (as was also shown by previous research) [26,27], and prior relevant research not being cited. While there were no significant differences between perceptions of authors, reviewers, and editors regarding the prevalence of undeserved authorship; editors perceived higher prevalence of relevant research not being cited as well as higher prevalences of fabrication, falsification, and plagiarism than authors or reviewers. These findings could be explained by the fact that most researchers often engage in conversations on authorship during their projects and publications, while less than 2% of researchers admitted to having fabricated or falsified data [28], or plagiarised other's work [29], and therefore the latter practices are more likely to have been encountered or discussed by editors than by other researchers. Additionally, we observed differences in perceived detrimental research practices, with younger researchers finding undeserved authorship, and not citing relevant research to be more prevalent than older researchers. This mimics previous studies which showed that young researchers often felt they were doing all of the work while others were receiving the credit, and that they had more research experience than many starting faculty members [24].

We also found that *Health Sciences* respondents perceived use of reporting guidelines, open peer review, publication of studies with null or negative results, and reporting of study limitations to be more prevalent than respondents from other disciplines, which is congruent with the frequency of recommendations on those topics in *Health Science* journals compared to other disciplines [15]. Finally, we also observed that researchers from USA and India perceived detrimental research practices to be more prevalent than respondents from other countries. This could be a consequence of the higher visibility of the Office of Research Integrity, and its legal foothold in the USA [30], as well as role of the Society for Scientific Values in India [31], the number of misconduct cases in those countries, as well as possibly higher competition levels in those countries [32,33].

While our study assessed attitudes and perceptions of a large number of respondents across many countries and disciplines, it is not without limitations. First, due to the low response rate our findings are not necessarily generalizable (we discuss possible influences of self-selection and non-response biases in detail in the Appendix). Our response rate, however, was similar to other recent large online surveys [16,22,34–36], and response rates have been consistently found to be lower in online versus other modes of survey dissemination [37]. The recent exception to this pattern was a 2019 survey of 2,801 researchers from economics, political science, psychology, and sociology regarding open science practices, which had a response rate of 46%. However, each participant in that survey was compensated with either 25 or 40 dollars (randomly) if they were students, or 75 or 100 dollars (randomly) if they were an author.

Second, while we did have respondents from 126 countries, we explored differences between respondents' attitudes and perceptions for only 4 countries with highest number of respondents in our survey. Further research is warranted to determine national or institutional differences [16,38]. Third, although our survey was confidential, previous research has suggested that researchers often overestimate detrimental research practices of their colleagues [28], but may also underreport such practices in order to protect the reputation of their field or themselves, for being unwilling to report such practices for official investigations [39]. Fourth, to preserve confidentiality, we did not ask information on respondents departments or universities. This precluded taking into account potential clustering of some observations as it is possible that witnessing the same detrimental practice or being aware of the same high-profile cases within a department or a field might have led to overestimation of such practices. Finally, we did not define all terms used in the survey, so some differences between respondents might also stem from their different interpretations of some terms. For example, previous surveys on falsification yielded higher estimates when the term 'falsified' was not used but researchers were instead asked if they had ever altered or modified their data [28]. Sixth, while we explored several sociodemographic characteristics, publication practices and knowledge of statistics as factors associated with respondents' attitudes and perceptions, previous research has also shown influence of respondents personality traits which we did not measure in our study [40].

In conclusion, our study has found that attitudes of authors, reviewers, and editors did not significantly differ regarding the TOP guidelines or their perceptions of their work climates. We also observed differences in the perceived prevalences of detrimental practices between editors and authors or reviewers, which highlights the need to raise awareness of these issues among all stakeholders, and to develop projects where all stakeholders would be working together to eradicate or minimize them. More studies are also needed to showcase the impact of any policy changes, as well as studies that lower the burden of implementing such policies [41]. Finally, recognition and rewarding of responsible practices should move from recommendations to actual practice [9].

## Supporting information

**S1 Appendix. Details on the survey's response rate, generalizability and bias, variable coding, and additional analyses are available in the appendix.**
(DOCX)

## Acknowledgments

### Declarations

We would like to thank both our pilot and survey respondents for their valuable inputs and taking the time to answer our questions. We would also like to apologise to them for the delay in publishing our results, which occurred in part due to MM moving to another institution and due to the COVID-19 pandemic and its effects on our personal and professional lives. MM is currently a postdoc at Meta Research Innovation Center at Stanford University, but as most of the study had been conducted during his postdoc in Amsterdam, the listed affiliation is the one that reflects the institute where most of the work was done. Finally, we would like to thank Ricardo Moreira, Research manager at Elsevier for scripting the survey and managing the fieldwork, and Robert Thibault for comments on the draft of our manuscript.

### Presentations at meetings/conferences

Preliminary results of our survey were presented at PUBMET 2018: The 5th Conference on Scholarly Publishing in the Context of Open Science held on September 20–21, 2018, in Zagreb, Croatia; as well as on the 6th World Conference on Research Integrity, held on June 2–5, 2019, in Hong Kong, China.

## Author Contributions

**Conceptualization:** Mario Maličhi, IJsbrand Jan Aalbersberg, Lex Bouter, Adrian Mulligan, Gerben ter Riet.

**Data curation:** Mario Maličhi.

**Formal analysis:** Mario Maličhi.

**Funding acquisition:** IJsbrand Jan Aalbersberg.

**Investigation:** IJsbrand Jan Aalbersberg, Lex Bouter, Adrian Mulligan, Gerben ter Riet.

**Methodology:** IJsbrand Jan Aalbersberg, Lex Bouter, Adrian Mulligan, Gerben ter Riet.

**Project administration:** IJsbrand Jan Aalbersberg, Adrian Mulligan.

**Resources:** IJsbrand Jan Aalbersberg, Lex Bouter, Gerben ter Riet.

**Software:** Adrian Mulligan.

**Supervision:** IJsbrand Jan Aalbersberg, Lex Bouter, Gerben ter Riet.

**Visualization:** Mario Maličhi, IJsbrand Jan Aalbersberg.

**Writing – original draft:** Mario Maličhi.

**Writing – review & editing:** Mario Maličhi, IJsbrand Jan Aalbersberg, Lex Bouter, Adrian Mulligan, Gerben ter Riet.

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
