## [Decision Letter · Decision Letter 0]

10 Mar 2022

PONE-D-22-03115Transparency in conducting and reporting research: a survey of authors, reviewers, and editors across scholarly disciplinesPLOS ONE

Dear Dr. Malički,

Thank you for submitting your manuscript to PLOS ONE. After careful consideration, we feel that it has merit but does not fully meet PLOS ONE’s publication criteria as it currently stands. Therefore, we invite you to submit a revised version of the manuscript that addresses the points raised during the review process.

 First of all, I would like to thank the 2 reviewers. Their comments were really helpful and fast. As you will see they consider that the manuscript is a robust one. Only minor changes were required. I have also a few minor comments too: - Please add a few words in your abstract about the main limitation in order to avoid any spin.- Please add in the text (not in the appendix), any change from the initial protocol. In my opinion it is important to be make sure that the reader can see these changes in a first look of the paper. I do think that all the suggestions by the reviewers will be easy to implement.

We look forward to receiving your revised manuscript.

Kind regards,

Florian Naudet, M.D., M.P.H., Ph.D.

Academic Editor

PLOS ONE

Journal Requirements:

"This study was part of an Elsevier funded project: Fostering Transparent and Responsible Conduct of Research: What can Journals do?. Details of the project are available on our project’s data repository: " ext-link-type="uri" xlink:type="simple">http://dx.doi.org/10.17632/53cskwwpdn.6.14"

"This study was part of an Elsevier funded project: Fostering Transparent and Responsible Conduct of Research: What can Journals do?. Details of the project are available on our project’s data repository: http://dx.doi.org/10.17632/53cskwwpdn.6. "

"IJsbrand Jan Aalbersberg is Senior Vice President of Research Integrity at Elsevier, and Adrian Mulligan is a Research Director for Customer Insights at Elsevier. Mario Malicki is a Co-Editor-In-Chief or Research Integrity and Peer Review journal. Other authors declare no competing interests."

Reviewers' comments:

Reviewer's Responses to Questions

**Comments to the Author**

1. Is the manuscript technically sound, and do the data support the conclusions?

Reviewer #1: Partly

Reviewer #2: Yes

2. Has the statistical analysis been performed appropriately and rigorously? 

Reviewer #1: I Don't Know

Reviewer #2: Yes

3. Have the authors made all data underlying the findings in their manuscript fully available?

Reviewer #1: Yes

Reviewer #2: Yes

4. Is the manuscript presented in an intelligible fashion and written in standard English?

Reviewer #1: Yes

Reviewer #2: Yes

5. Review Comments to the Author

Reviewer #1: I am slightly concerned the the conclusion, whilst I agree with it, doesn't follow on logically from the survey results which may or may not be representative of reality anyway. We can all agree that policies to foster better research practices should be enforced rather than encouraged, but it is a bit of a leap to conclude this from the results of a survey showing little difference in attitudes and experience to these standards between three groups of stakeholders.

Reviewer #2: I uploaded the review that I copied below:

This article is interesting, of good quality and deserves publication subject to some clarification. The Appendix is worth reading.

Comments on a few points

• Introduction (page 3) is well done and sets out the problem. In the first sentence, it would be useful to clarify that the 3 million articles per year data is an estimate of the STM segment, and that there is no HHS data for the number of articles.

• Methods: Correctly stated, given that the study protocol is available. Perhaps add, if available, the estimated time to answer all questions.

• Results (page 6). The population needs to be detailed a bit more, and it's all in the appendix. It must be said that there were 28 fields for the disciplines, and especially cite the proportion for health sciences, the results of which appear below. Further on, the data analysed concern health sciences (xx%) which could be confused with life sciences (25%)

• No comments on all the tables: they are well done; the statistical tests are in the appendix and this is sufficient; perhaps some comments in the text could be reduced as redundant, but it is not essential

• Discussion (page 13 and following). The discussion needs to be reconsidered to address a few points.

• Would it be useful to compare your 4.9% response rate with E Fong's 10.5% response rate on a sample of 110,000 emails. Why such a difference? I haven't looked closely at the E Fong article, and I don't know if it's relevant. https://journals.plos.org/plosone/article?id=10.1371/journal.pone.0187394

• The distinction between authors and reviewers is rather artificial as all reviewers are authors. The comparisons of authors / reviewers / editors are disappointing. Should table A1 be discussed?

• It is a pity that the notion of gender has not been explored. These are points for discussion.

• Sometimes the discussion repeats data from the tables, and this could reduce the length a bit; the conclusion is not useful

• I agree with the sentence on page 13 'Direct comparisons of our results with other surveys are difficult'... but the discussion is just that, when there are issues that deserve discussion:

o The discussion is mainly focused on researchers and very little on journals and peer review;

o Wouldn't it make sense to discuss the responsibility of journals? This is a huge topic, but isn't it the major point for improving the system. How can TOP be implemented more quickly? Better recognition of peer review, etc. is essential to develop TOP. What can journals do to implement the TOP guidelines faster?

o Why not compare the implementation of TOPs, guidelines to the implementation of protocol registration: the ICMJE requested protocol registration in 2004, and almost 20 years later, only prestigious journals require and control this registration.

6. PLOS authors have the option to publish the peer review history of their article (what does this mean?). If published, this will include your full peer review and any attached files.

Reviewer #1: No

Reviewer #2: **Yes: **Hervé Maisonneuve

---

## [Author Response · Author response to Decision Letter 0]

2 Jun 2022

Dear Editor Florian Naudet,

Thank you and the reviewers for your comments and kind words for our study. We present below a point-by-point response for all style requirements and review suggestions. Sentences in red are those that have been added during the revision.

Reply: The style changes have been applied. We did not use the track changes for this, as they would have obscured the replies to reviewer comments. 

Reply: The references have been reviewed and formatted to PLOS style. The only reference change was updating our old reference 37 which was a preprint to its peer reviewed version (https://doi.org/10.1101/2020.11.25.395376 to https://doi.org/10.1371/journal.pone.0244529 ). As an editor, I would also like to praise your rules on indicating the article retracted status – however I would recommend you have a guide to authors on how to check this, and provide them with a tool that does this for their submitted manuscripts. For this manuscript we checked the retracted status using Zotero. None of our references refer to retracted studies.

"This study was part of an Elsevier funded project: Fostering Transparent and Responsible Conduct of Research: What can Journals do? Details of the project are available on our project’s data repository: http://dx.doi.org/10.17632/53cskwwpdn.6.14"

"This study was part of an Elsevier funded project: Fostering Transparent and Responsible Conduct of Research: What can Journals do? Details of the project are available on our project’s data repository: http://dx.doi.org/10.17632/53cskwwpdn.6. "

Reply: The funding section has been removed from the acknowledgments, the text to be used is listed now in our new cover letter. 

"IJsbrand Jan Aalbersberg is Senior Vice President of Research Integrity at Elsevier, and Adrian Mulligan is a Research Director for Customer Insights at Elsevier. Mario Malicki is a Co-Editor-In-Chief or Research Integrity and Peer Review journal. Other authors declare no competing interests."

Reply: The Competing Interests section has been removed from the acknowledgments, the text to be used is listed now in our cover letter and includes the sentence: This does not alter our adherence to PLOS ONE policies on sharing data and materials. 

Editor’s comments:

1. First of all, I would like to thank the 2 reviewers. Their comments were really helpful and fast. As you will see they consider that the manuscript is a robust one. Only minor changes were required. I have also a few minor comments too: 

Please add a few words in your abstract about the main limitation in order to avoid any spin.

Reply: The following sentence has been added to the abstract: While survey respondents came from 126 different countries, due to the survey’s overall low response rate our results might not necessarily be generalizable.

2. Please add in the text (not in the appendix), any change from the initial protocol. In my opinion it is important to be make sure that the reader can see these changes in a first look of the paper. I do think that all the suggestions by the reviewers will be easy to implement.

Reply: The changes from the protocol have been moved from the appendix to the main manuscript. Their full added text is: Due to miscommunication within the team, instead of adding the collected email addresses of the editors from our previous study on Instruction to Authors to the total of 100,000 planned invites, 58 of collected email addresses were used for piloting the survey, and 292 were incorporated into the 100,000 invites sent. Additionally, during survey creation, instead of planned options of 0, 1-4, 5-10,10+ for question - How many articles (approximately) did you review in the last 12 months? survey options were instead 1, 2-5, 6-10, and 10+.

Reply: All of the above have been submitted and accordingly labeled. 

Reviewers' comments:

Reviewer #1: 

1. I am slightly concerned the conclusion, whilst I agree with it, doesn't follow on logically from the survey results which may or may not be representative of reality anyway. We can all agree that policies to foster better research practices should be enforced rather than encouraged, but it is a bit of a leap to conclude this from the results of a survey showing little difference in attitudes and experience to these standards between three groups of stakeholders.

Reply: We thank the reviewer for this comment, and even though we also personally agree about enforcements, we have to emphasize that we did not state anywhere in our conclusions that any of the practices should be enforced. We only stated that our results showed a mismatch between the practices of researchers and current recommendations, and that due to this “greater involvement of all stakeholders is needed to align actual practices with current recommendations.” There are multiple ways how this could be achieved, including education and awareness raising, or changing of incentives for hiring, promotion and tenure and these have been stated in the discussion as: which highlights the need to raise awareness of these issues among all stakeholders, and to develop projects where all stakeholders would be working together to eradicate or minimize them. If the reviewer is referring to our last sentence: which states recognition and rewarding of responsible practices needs to move from recommendations to actual practice, we changed the need to should in the revised manuscript so that it does not confuse the readers that this means enforcement. This sentence was however a reflection of the fact that DORA declaration and Hong Kong principles for assessment of researcher were endorsed by only a small fraction of institutions, i.e., there are more than 32 000 universities in the world and only 2560 institutions (8%) endorsed DORA. 

Reviewer #2:

1. This article is interesting, of good quality and deserves publication subject to some clarification. The Appendix is worth reading.

Reply: We thank the reviewer for these kind words. 

2. Comments on a few points

• Introduction (page 3) is well done and sets out the problem. In the first sentence, it would be useful to clarify that the 3 million articles per year data is an estimate of the STM segment, and that there is no HHS data for the number of articles.

Reply: We have modified the sentence to state with estimates of more than 3 million articles published per year. The reviewer might also be interested that field distribution of published articles can be seen using Dimension.AI or Open Alex databases (e.g. https://app.dimensions.ai/analytics/publication/for/aggregated?or_facet_publication_type=articleor_facet_year=2021and_facet_publication_type=article )

3. Methods: Correctly stated, given that the study protocol is available. Perhaps add, if available, the estimated time to answer all questions.

Reply: The following sentence was added to the methods: Estimated time to finish the survey was 12 minutes (based on pilot results), and was listed in the survey invite letter. 

4. Results (page 6). The population needs to be detailed a bit more, and it's all in the appendix. It must be said that there were 28 fields for the disciplines, and especially cite the proportion for health sciences, the results of which appear below. Further on, the data analysed concern health sciences (xx%) which could be confused with life sciences (25%)

Reply: We thank the reviewer for this comment. It is always a delicate balance to decide which information to keep in the appendix and which in the main manuscript. The information on disciplines has now been added to the methods, and percentage for health sciences listed in results. While some readers may be confused between life and health science, it was based on Scopus journal classification from which we sampled participants, and it the same classification we used in our two other papers (https://doi.org/10.1371/journal.pone.0222157, https://www.nature.com/articles/s41467-021-26027-y) that were from the same project. The methods text now states: In section 4, respondents could select one of 28 prespecified scholarly fields, a multidisciplinary category or the “other” category where they could write their field(s) using an open text response format. All answers they provided in free text, as well as the 29 choices, were recoded to one of the 6 major categories that were used in our previous study on Instruction to Authors: Arts Humanities, Health Sciences, Life Sciences, Physical Sciences, Social Sciences, and Multidisciplinary.[15]. And the added results text states: Health Sciences (13%). 

5. No comments on all the tables: they are well done; the statistical tests are in the appendix and this is sufficient; perhaps some comments in the text could be reduced as redundant, but it is not essential.

Reply: We thank the reviewer for the kind words, we kept the result text as is, as we feel it is still quite a shortened version with many additional results in the appendix.

6. Discussion (page 13 and following). The discussion needs to be reconsidered to address a few points: Would it be useful to compare your 4.9% response rate with E Fong's 10.5% response rate on a sample of 110,000 emails. Why such a difference? I haven't looked closely at the E Fong article, and I don't know if it's relevant. https://journals.plos.org/plosone/article?id=10.1371/journal.pone.0187394

Reply: The Fong study is an interesting one, however due to following reasons we do not feel it is a good comparison to our study: 1) the Fong study are actually 4 surveys done over a period of 5 years, and it employed a different sampling strategy – emails from respondents were obtained through associations (e.g., American Economic Association), annual meeting lists, and from University websites – this makes it more likely that those were active emails those individuals used, while in our study we used corresponding emails of papers, and we know those change when researcher move from one university to another. That is why we compared ours to other surveys who used corresponding emails. 2) Their survey was shorter – and this also likely affects response rates. 3) Their sampling included only American researchers, and was funded and supported by ORI - Office of Research Integrity – an important and a very well known institution for Research Integrity in USA. We on the other hand sampled researchers from across the world, and our project was funded by Elsevier - information we disclosed in the invite letter – and these two differences may also have impacted response rates. 

7. The distinction between authors and reviewers is rather artificial as all reviewers are authors. The comparisons of authors / reviewers / editors are disappointing. Should table A1 be discussed?

Reply: While it is true that (almost all, 93%) reviewers are also authors, the reverse does not apply. This is reflected in the differences in age and number of publications between authors and reviewers in our study. One could say that reviewers are essentially more experienced authors, and while we did not collect information about the number of different journals they reviewed for in their life time, we can presume that with each review a reviewer learns about the practices of other journals, their review forms or instructions, and they may learn from comments of other reviewers. And so, it can be reasonable to expect that review experience might lead to differences in attitudes towards transparency, in their perceptions of prevalence of misconduct and even in the quality of their work. Detectable differences in our outcomes could have been found if we looked at only the most experienced reviewers in our study, or may be found in other studies that might focus on those that produce more than 100 or even 500 reviews per year 

(https://publons.com/researcher/?is_core_collection=1is_last_twelve_months=1order_by=num_reviews). However, as researchers, we must also be consistent to our planned study protocol. Our objective was to explore differences between authors, reviewers, and editors - and use their self-reported role(s) in Q1 of our survey to group them in these roles. Our shared data allow researchers to group participants differently and conduct sub-analyses. In light of this, we don’t think additional discussion is needed regarding Table 1, and to limit an already long discussion, we would rather expand on the other issues the reviewer raised below. 

8. It is a pity that the notion of gender has not been explored. These are points for discussion.

Reply: We did explore differences between male and female respondents, as can be seen in the Appendix Tables 7 to 9 but found no statistically significant differences. We chose not to focus on this in discussion, due to several reasons. One, it is difficult to find a survey with which to compare our results. Some surveys of specific countries have found gender differences between data sharing attitudes (e.g., Croatia https://doi.org/10.1371/journal.pone.0244529, or Germany https://doi.org/10.1371/journal.pone.0183216), but due to different sampling methods, how the questions were asked, response rates, etc., we feel these are not appropriate comparisons. Additionally, we wanted to focus in the discussion on the observed differences for our main objective – between authors, reviewers and editors. Our survey was quite large, and it is hard to comment on each of the 38 questions in regards to gender as they could all be individual topics, or questions asked one day in sys. reviews. We are of course willing to expand on the gender in the discussion if there is something very specific the review or editor wants us to mention, or if they had a specific study as comparison in mind.

9. Sometimes the discussion repeats data from the tables, and this could reduce the length a bit; the conclusion is not useful

Reply: We have intentionally repeated data for 11 out of 38 questions our survey addressed, as we wanted to spare the readers of going back to the tables while they read the discussion on these specific topics. We are open to removing some of them if the editor or the reviewer insist, but we do not think that much is gained by shortening the discussion by removing 5-6 numbers/words. And while we could cut more words by cutting some of the 11 out of the 38 topics we discuss in the discussion, we prefer to keep these 11, as we still omitted many others to be mindful of the length of the discussion. As for the conclusions, we hope that added texts written below make the conclusion and discussion more useful to the readers and the reviewer. 

10. I agree with the sentence on page 13 'Direct comparisons of our results with other surveys are difficult'... but the discussion is just that, when there are issues that deserve discussion:

o The discussion is mainly focused on researchers and very little on journals and peer review;

o Wouldn't it make sense to discuss the responsibility of journals? This is a huge topic, but isn't it the major point for improving the system. How can TOP be implemented more quickly? Better recognition of peer review, etc. is essential to develop TOP. What can journals do to implement the TOP guidelines faster?

o Why not compare the implementation of TOPs, guidelines to the implementation of protocol registration: the ICMJE requested protocol registration in 2004, and almost 20 years later, only prestigious journals require and control this registration.

Reply: We agree with the reviewer that we could have given more discussion in this paper to the role of journals and peer review – but as we discussed those roles quite extensively in two other publications that came from our project Fostering Transparent and Responsible Conduct of Research: What can Journals do? - of which this survey was also a part, and as in those two publications we discussed ICMJE, protocol registrations, as well as reporting guidelines, we did not want to repeat the same or similar messages. We also planned to summarize all publications from this project in a report that would focus on exactly the action that journals should take. Nevertheless, the reviewer is right to ask this, and so we added the following to the discussion: It is likely that the lack of support of editors in our study towards all aspects of TOP recommendations is one of the contributing factors why these recommendations have not been endorsed by a larger number of journals today. The roles of editors and journals in endorsing or requiring specific practices, the lack of resources they often face, and the lack of proper intervention studies, were discussed extensively in our previous publications. [15, 23] Here we would like to add that uptake and attitudes toward different practices are also likely influenced by whom and how recommendations are made. More rigorous methodological steps during recommendation development (similar to those for reporting guidelines or clinical practice guidelines), and open feedback calls might have led to higher initial uptakes. Additionally, published case reports with cost estimates and practical tips from those involved in recommendation development, that then managed to change practices of their journals, departments or institutions, could perhaps lead to additional endorsements. 

Kind regards, 

in the name of all co-authors,

Mario Malički

---

## [Editor Report · Decision Letter 1]

3 Jun 2022

Transparency in conducting and reporting research: a survey of authors, reviewers, and editors across scholarly disciplines

PONE-D-22-03115R1

Dear Dr. Malički,

We’re pleased to inform you that your manuscript has been judged scientifically suitable for publication and will be formally accepted for publication once it meets all outstanding technical requirements.

Kind regards,

Florian Naudet, M.D., M.P.H., Ph.D.

Academic Editor

PLOS ONE

Additional Editor Comments (optional):

Thank you for the revisions and kudos for this important paper. As those were minor edits, I have assessed all your answers and will ask PLOS One to FWD your answers to the reviewers. Best.
---

## [Editor Report · Acceptance letter]

7 Jul 2022

PONE-D-22-03115R1 

Transparency in conducting and reporting research: a survey of authors, reviewers, and editors across scholarly disciplines 

Dear Dr. Malički:

I'm pleased to inform you that your manuscript has been deemed suitable for publication in PLOS ONE. Congratulations! Your manuscript is now with our production department. 

Kind regards, 

on behalf of

Pr. Florian Naudet 

Academic Editor

PLOS ONE